∂ | Open Peer Review | Bacteriology | Methods and Protocols

# Massive culture-based approach for the screening of AmpC, ESBL, and carbapenemase producers from rectal swabs

Gabriel Taddeucci-Rocha,[1,2] Victoria de Oliveira Costa,[1] Sarah Vitória Martins da Silva,[1] Jéssica Britto Gonçalves,[1] Natalia Chilinque Zambão da Silva,[3,4] Marcia Maiolino Garnica,[4,5] Renata Cristina Picão[1,6]

**ABSTRACT** Gram-negative bacilli-producing beta-lactamases are major causes of difficult-to-treat infections, especially the AmpC, extended-spectrum beta-lactamases (ESBL), and carbapenemase types. Their spread within and outside hospital settings demands effective detection and monitoring in various environments, but current methods for this purpose often neglect important groups of beta-lactamases or are expensive and time consuming. We aimed to develop and test a massive culture approach to detect and differentiate between beta-lactamase producers from complex samples. The method includes enrichment on MacConkey agar supplemented with ceftriaxone to select for AmpC, ESBL, and carbapenemase producers, followed by replica plating under selective pressures (cefoxitin, cefepime, and imipenem) to differentiate them. The massive culture approach effectively differentiated strains producing different beta-lactamases in mixed cultures. In tests with rectal swabs, our method demonstrated 100% sensitivity, higher specificity, and greater accuracy for ESBL detection compared to the reference method. Additionally, it identified a broader spectrum of beta-lactamase producers, including AmpC and carbapenemase. The massive culture approach is a promising tool for detecting and differentiating gram-negative bacilli-producing beta-lactamases from rectal swabs. Due to the additional time required to produce results, this method is most suitable for central and research laboratories and enhances surveillance capabilities for antimicrobial resistance.

**IMPORTANCE** The intestinal tract is a major reservoir of multidrug-resistant gram-negative bacilli, and surveillance of colonization is essential to understand resistance dissemination in both community and healthcare settings. However, standard culture-based methods typically target specific resistance mechanisms, often overlooking the coexistence of distinct beta-lactamase-producing strains within a single host. This limits our understanding of colonization dynamics and resistance evolution. To address this gap, we developed a culture-based approach that assesses the growth of strains under different selective pressures from a single rectal swab. By combining enrichment with replica plating, our method enables phenotypic discrimination between AmpC-, extended-spectrum beta-lactamases-, and carbapenemase-producing bacteria without depleting sample material across multiple media. Although validated for rectal swabs, the approach may be adapted for other complex samples such as urine, blood, soil, or water, expanding its utility in diverse clinical and environmental investigations. This strategy enhances the detection of diverse resistance profiles and supports a more comprehensive view of colonization and antimicrobial resistance dynamics.

**KEYWORDS** replica plating, detection, antimicrobial resistance, surveillance studies, rectal swab

**Peer Reviewers** Vinay Modgil, HISP India, Chandigarh, India; Neha Khaware, Indian Institute of Technology Delhi, New Delhi, India

Address correspondence to Renata Cristina Picão, renata.picao@micro.ufrj.br; renata@ufscar.br.

The authors declare no conflict of interest.

See the funding table on p. 10.

10.1128/spectrum.00157-25 **1**

Beta-lactam-resistant gram-negative bacilli are major causes of infections in both hospitals and community settings, primarily due to the production of beta-lactamases (1). These enzymes include AmpC-like cephalosporinases, extended-spectrum beta-lactamases (ESBL), and carbapenemases, which confer resistance to various beta-lactams (2). Genes encoding these beta-lactamases are often carried along with resistance determinants against other antimicrobial classes, severely limiting therapeutic options (3). For these reasons, beta-lactamase producers are key targets for antimicrobial resistance (AMR) research (4).

The spread of beta-lactamase producers outside hospitals is a growing global concern with unknown consequences for human, animal, and environmental health (5). Thus, monitoring their presence in hospitals is crucial for infection control, while surveillance of carriage in asymptomatic humans, animals, and environments is essential for risk assessment and understanding the environmental spread of AMR (6).

Surveillance of these microorganisms among humans typically involves culturing fecal specimens under selective pressure to target specific mechanisms such as ESBL or carbapenemase production, followed by analysis of isolated colonies. Molecular detection methods for beta-lactamase resistance mechanisms exist but often focus on a few known carbapenemases, neglecting acquired ESBLs, AmpCs, some carbapenemases, and undescribed beta-lactamase families that may also threaten anti-infective therapy (7) or involve sequencing of the entire resistome using metagenomics (8). The latter requires specialized equipment and highly trained personnel in bioinformatics, limiting its use for routine diagnostic and research purposes, especially in low- and middle-income countries (9). Consequently, phenotypic investigation through selective culture remains the reference method to survey the carriage of beta-lactamase producers. However, colonization may involve different strains carrying various AMR mechanisms, so the targeted culture-based approach may limit the understanding of strains and mechanisms driving the spread of beta-lactam resistance in different settings (10).

Replica plating is a classical method that allows the evaluation of all colonies present in a culture and enables assessment of their growth under various selective pressures (11). We hypothesized that a massive culture approach based on replica plating is suitable for detecting and differentiating beta-lactamase producers from complex samples. Our study aims to (i) standardize a massive culture approach for the identification of AmpC, ESBL, and carbapenemase producers; (ii) evaluate its performance in detecting ESBL producers in rectal swabs; and (iii) describe the detection of plasmid-mediated AmpC (pAmpC) and/or carbapenemases in rectal swabs using this approach.

## MATERIALS AND METHODS

### Protocol standardization

The approach we envisioned is based on an enrichment culture on MacConkey agar (Difco, Michigan, USA) supplemented with ceftriaxone (Agila, Bangalore, India), followed by replica plating under selective pressures using cefoxitin, cefepime, and imipenem (all Sigma-Aldrich, Missouri, USA) to differentiate between AmpC, ESBL, and carbapenemase producers, respectively (Fig. S1).

Initially, we aimed to identify the ideal drug concentration for replica plating by testing three gram-negative bacilli that produce different beta-lactamases (pAmpC, ESBL, and carbapenemase) and exhibit distinct colony characteristics on MacConkey agar, along with negative control. These included ESBL-producing *Klebsiella pneumoniae* 134 (CTX-M-15, large, mucoid, lactose-fermenting colonies) (12), plasmid-mediated AmpC-producing *Escherichia coli* D493 trans-III (CMY-2, medium-sized opaque lactose-fermenting colonies) (13), and carbapenemase-producing *Pseudomonas aeruginosa* C237 (VIM-2, medium-sized flat colonies with irregular margins, lactose-non-fermenting) (14), as well as the negative control *E. coli* TOP10 (small lactose-non-fermenting colonies).

An initial inoculum of $10^8$ CFU/mL was prepared for each bacterium using the 0.5 McFarland standard as a reference. Equal volumes of each suspension were combined in a single 1.5 mL tube and diluted to reach $10^5$ CFU/mL and $10^4$ CFU/mL. Fifty-microliter aliquots of the diluted mixed suspensions were seeded, in triplicate, onto antimicrobial-free MacConkey agar plates without antimicrobials (MAC) and onto MAC supplemented with ceftriaxone (MCRO, 1.5 µg/mL), then incubated overnight at 35°C ± 2°C under aerobic conditions. The microbial growth on each plate was replicated onto eight MAC: one initial and one final plate without antimicrobials, for inoculum control, and two plates with each drug concentration (1× and 2.5× the breakpoint for resistance according to the Clinical and Laboratory Standards Institute [CLSI] M100-Ed33 document). These included cefepime (FEP, 16 and 40 µg/mL), cefoxitin (FOX, 32 and 80 µg/mL), and imipenem (IPM, 4 and 10 µg/mL).

Sterile 15 × 15 cm pieces of 2.0 mm thick elastic velvet were fixed to an 89 mm diameter support made of autoclavable resin and printed on an stereolithography (SLA) 3D printer. Velvet sterile control was assessed by pressing a tryptic soy agar (TSA; Difco) plate against the fabric. Fresh cultures from MCRO were transferred to the velvet by gently pressing the culture against it, followed by replication on the series of plates by gently pressing each plate against the velvet. All culture plates were incubated overnight at 35°C ± 2°C under aerobiosis. The parameter used to choose the best concentration for further experiments was the observation of well-formed colonies corresponding to the expected microorganism(s). On plates supplemented with cefepime, it should only grow those strains producing ESBL and carbapenemase; on plates supplemented with cefoxitin, it should only grow strains producing AmpC and carbapenemase; and on plates supplemented with imipenem, only the growth of strains producing carbapenemase was expected (Fig. S1).

We then assessed the performance of the massive culture approach in a non-controlled sample. For this, we seeded a rectal swab provided by a volunteer in MCRO and replicated the culture onto five MacConkey agar plates: without supplementation, with cefepime (MFEP, 16 µg/mL), with cefoxitin (MFOX, 32 µg/mL), with imipenem (MIPM, 4 µg/mL), and again without supplementation. The detailed protocol is available (Supplemental material and Supplemental video). Colonies observed on MFOX, MFEP, and MIPM were investigated for plasmid-mediated AmpC (pAmpC), ESBL, and carbapenemase production, respectively, as described below.

## Accuracy of the massive culture approach to detect ESBL producers in rectal swabs

We evaluated the performance of the proposed method for detecting ESBL in self-collected rectal swabs, comparing it to the performance of a commercial chromogenic agar (Laborclin, Paraná, Brazil), hereafter referred to as CESBL. The fecal specimens were provided by 30 volunteers, who were recruited during a 3-month collection period established for the study. The swabs were first spotted in TSA to verify the presence of microorganisms, ensuring that the specimens were properly collected. They were then cultured on CESBL and MCRO and incubated aerobically for 18–24 hours at 35°C ± 2°C. Specimens showing no bacterial growth on TSA were excluded from further analysis. To avoid biased results due to insufficient fecal material, we first cultured 15 swabs on CESBL and then on MCRO, while another 15 swabs were first inoculated on MCRO and then on CESBL.

MCRO plates showing bacterial growth were replicated onto five MacConkey agar plates: MAC, MFOX, MFEP, MIPM and MAC. For the specific purpose of evaluating the accuracy of the massive culture approach in detecting ESBL producers, only colonies grown on CESBL and MFEP were further studied. Up to five representatives of each colony morphotype were selected and identified using Matrix-assisted laser desorption ionization - time of flight (MALDI-TOF) MS Microflex LT and the Biotyper 3.0 software, according to the manufacturer's instructions (Bruker Daltonics, Bremen, Germany). Enterobacteria were evaluated for ESBL production by the double

disk synergy test, and positive isolates were confirmed by PCR targeting $bla_{CTX-M-1/2-like}$, $bla_{CTX-M-8-like}$, $bla_{CTX-M-9-like}$, $bla_{SHV-like}$, $bla_{TEM-like}$, and $bla_{GES-like}$ genes. For the $bla_{CTX-M}$ group, primers mCTX 1/2-F (ATGTGCAGYACCAGTAA) and mCTX 1/2-R (CGCTGCCGGTTTTATCSCCC) amplified a 512 bp product for groups 1 and 2. For group 8, mCTX 8-F (AACRCRCAGACGCTCTAC) and mCTX 8-R (TCGAGCCGGAASGTGTYAT) produced a 333 bp fragment. For group 9, mCTX 14-F (GGTGACAAGAGARTGCAACGGAT) and mCTX 14-R (TTACAGCCCTTCGGCGATGA) amplified an 876 bp product. Thermocycling included initial denaturation at 94°C for 3 minutes, 35 cycles of 94°C for 30 seconds, 51°C for 60 seconds, and 72°C for 2 seconds, with a final extension at 72°C for 5 minutes. For other genes, mTEM-F (CCCTTATTCCCCTTTYTGCGG) and mTEM-R (AACCAGCCAGCCWGAAGG) amplified a 650 bp fragment of $bla_{TEM}$. For $bla_{GES}$, mGES-F (AACCAGCCAGCCWGAAGG) and mGES-R (CCGTGCTCAGGATGAGTTG) produced a 750 bp product. For $bla_{SHV}$, mSHV-F (CTTGACCGCTGGAAAACGG) and mSHV-R (AGCAACGGAGC GATCAACGG) resulted in a 200 bp fragment. Thermocycling included initial denaturation at 94°C for 5 minutes, 30 cycles of 95°C for 30 seconds, 54°C for 20 seconds, and 72°C for 30 seconds, with a final extension at 72°C for 10 minutes. *Pseudomonas* spp. and *Acinetobacter* spp. were only investigated for the presence of ESBL-encoding genes due to the poor performance of the phenotypic test in detecting ESBL production in these microorganisms. Isolates, where ESBL production is clinically irrelevant, were not investigated for this resistance mechanism.

The growth of colonies with confirmed ESBL production was considered a true-positive result for both CESBL and the massive culture approach on MFEP. False positives were defined as the growth of colonies without such confirmation. True-negative results were recorded when no colonies grew on selective media for both tests, and false negatives were noted when there was no growth in one technique but a true-positive result in the other. Sensitivity, specificity, positive predictive values (PPV), and negative predictive values (NPV) were calculated to compare the performance of the massive culture approach and the reference method in detecting ESBL producers in rectal swabs (Table 1).

## Detection of pAmpC and carbapenemase producers using the massive culture approach

We additionally studied the colonies grown under cefoxitin and imipenem selective pressures from rectal swabs with Cary-Blair transport medium (FirstLab, Paraná, Brazil) to assess the occurrence of AmpC and carbapenemase producers, respectively. Up to five representatives of each colony morphotype were subcultured under the same conditions and identified using MALDI-TOF. Production of pAmpC was assessed among *E. coli* and *Klebsiella* spp., which are typically susceptible to third-generation cephalosporins due to low expression or absence of intrinsic AmpC-like cephalosporinases, respectively.

**TABLE 1** Formulas and performance of the massive culture approach (MFEP) and the targeted approach (CESBL) in detecting ESBL producers[a,b]

| | Selective pressure | | |
|---|---|---|---|
| Parameters | CESBL | MFEP | *P* value |
| TP | 10 | 10 | — |
| FP | 12 | 8 | — |
| TN | 3 | 7 | — |
| FN | 0 | 0 | — |
| Sensitivity TP/(TP + FN) | 1.00 (0.72–1.00) | 1.00 (0.72–1.00) | — |
| Specificity TN/(TN + FP) | 0.20 (0.07–0.45) | 0.47 (0.25–0.70) | 0.121 |
| PPV TP/(TP + FP) | 0.45 (0.27–0.65) | 0.56 (0.34–0.75) | 0.525 |
| NPV TN/(TN + FN) | 1.00 (0.44–1.00) | 1.00 (0.65–1.00) | — |
| Accuracy (TP + TN)/(TP + FP + TN + FN) | 0.52 (0.33–0.70) | 0.68 (0.48–0.83) | 0.248 |

[a]TP, true positive; FP, false positive; TN, true negative; FN, false negative.
[b]"—", not applicable.

Isolates resistant to cefoxitin and amoxicillin/clavulanate, assessed by CLSI disk diffusion (15), were subjected to PCR targeting genes encoding MOX, CIT, DHA, ACC, EBC, and FOX pAmpC groups, as previously described (16), followed by amplicon sequencing. Carbapenemase production was evaluated using the modified carbapenem inactivation method (17, 18) and the EDTA-modified carbapenem inactivation method (19), and confirmed by the detection of carbapenemase-encoding genes (20).

## Statistical analysis

Diagnostic performance measures (sensitivity, specificity, PPV, NPV, and accuracy) were calculated based on $2 \times 2$ contingency tables constructed for each selective pressure evaluated. The 95% confidence intervals were calculated using the exact Clopper–Pearson method. Comparisons between proportions were performed using Fisher's exact test, considering a significance level of 5% ($P < 0.05$).

## RESULTS AND DISCUSSION

The massive culture approach based on replica plating under selective pressure, designed to detect AmpC, ESBL, and carbapenemase producers in rectal swabs effectively differentiated bacteria producing specific beta-lactamases even when outnumbered by other colonies. Its application to rectal swabs demonstrated greater specificity for detecting ESBL producers compared to conventional targeted methods, with the additional benefit of simultaneously identifying AmpC and carbapenemase-producing isolates.

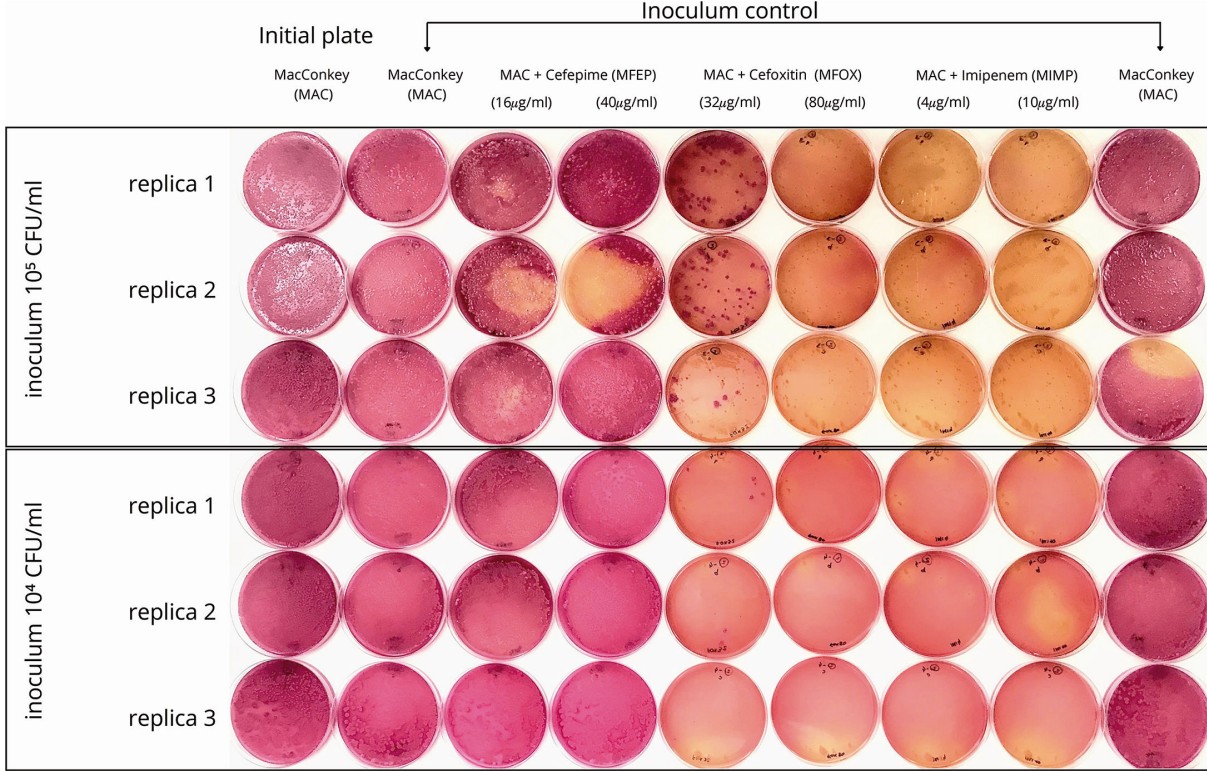

**FIG 1**  Result of massive culture using replica plating for different CFU per milliliter counts of a mixed suspension, including previously characterized strains producing pAmpC, ESBL, carbapenemase, and a negative control (*E. coli* TOP10). The suspensions were initially cultured overnight on MacConkey agar without supplementation (MAC), and then replicated onto: a MAC plate for initial inoculum control; cefoxitin (MFOX 32 and 80 µg/mL), cefepime (MFEP 16 and 40 µg/mL), imipenem (MIMP 4 and 10 µg/mL), and a MAC plate for final inoculum control. Growth of all strains is expected in MAC plates, while the CTX-M-15-producing *K. pneumoniae* (forming large, mucoid, and lactose-fermenting colonies) is expected to grow in MFEP; CMY-2-producing *E. coli is* expected to grow on MFOX (forming medium-sized, opaque, lactose-fermenting colonies); and the VIM-2-producing *P. aeruginosa* is expected to grow on MFOX, MFEP, and MIMP (forming medium-sized flat colonies with irregular margins, lactose-non-fermenting).

We first optimized the protocol by determining the antimicrobial concentrations that yielded well-formed colonies of target microorganisms, using well-characterized strains. Although not all colonies were visible in Fig. 1, the observed growth patterns shifted clearly in response to selective pressures. The lowest tested concentrations of each antimicrobial agent allowed exclusive growth of target strains, whereas higher concentrations caused partial inhibition, particularly evident with cefoxitin. Final inoculum control plates retained comparable growth to the initial plates, confirming that reduced growth was attributable to antimicrobial activity rather than inoculum variation. Of note, we were able to detect resistant strains even at low inoculum sizes and minimal drug concentrations, despite being masked by other colonies in non-selective plates. This demonstrates the approach's sensitivity under real-world conditions, where target strains are often present in lower abundance. Based on these observations, we selected the lowest effective antimicrobial concentrations for the subsequent screening of beta-lactamase producers.

When applied to rectal swabs, the method successfully differentiated beta-lactamase producers regardless of their relative abundance. Figure 2 illustrates the culture results from the replication of a rectal swab plated on MCRO. Most colonies observed in control plates also appeared on MFOX, suggesting the presence of AmpC producers. Notably, three colonies suspected to be ESBL producers were observed on MFEP, while complete growth inhibition on MIPM indicated the absence of carbapenemase producers. Eight isolates from MFOX were identified as *Citrobacter* spp. (4) and *Pseudomonas* spp. (4), both intrinsic AmpC producers. The two colonies grown on MFEP were identified as *E. coli* and confirmed as ESBL producers by phenotypic testing and PCR and amplicon sequencing, which revealed 100% identity to $bla_{CTX-M-79}$ and $bla_{CTX-M-55}$ genes (839 bp amplicon).

To evaluate diagnostic performance, we compared the massive culture method to a commercial chromogenic ESBL screening agar (CESBL) using 25 rectal swabs (five were excluded due to the absence of growth on TSA). Both methods detected ESBL-producing strains in 10 specimens. However, the bacterial identification and resistance gene profiles showed partial discrepancies between the isolates recovered on CESBL and those from MFEP plates (Table 2). Among the 159 total isolates (71% Enterobacteriaceae, 28% non-fermenting gram-negative bacilli [NFGNB], and 1% gram-positive cocci), only Enterobacteriaceae were confirmed as ESBL producers. MFEP replica plating yielded a

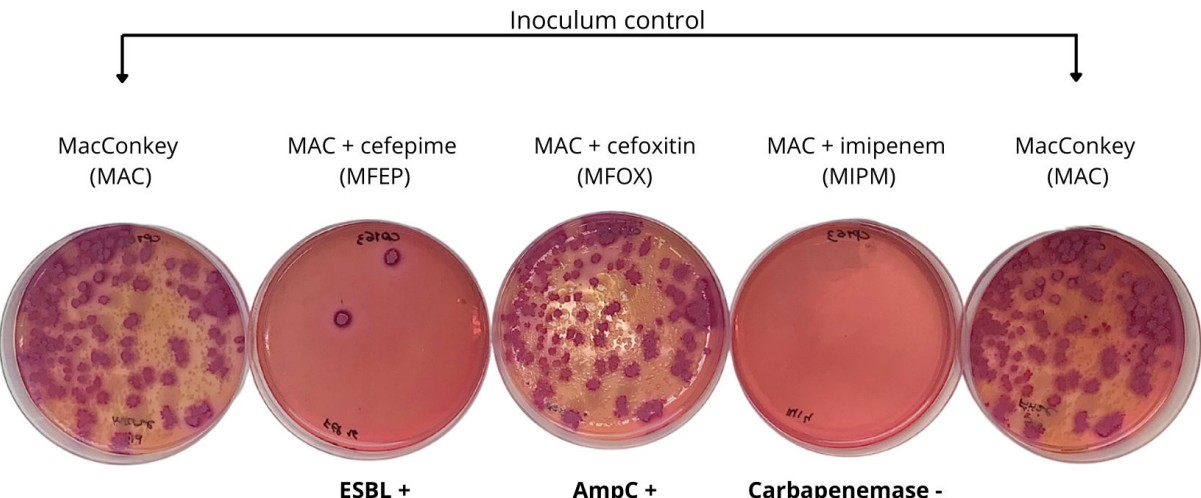

**FIG 2**  Replication of growth observed after seeding the rectal swab on MacConkey agar plate supplemented with ceftriaxone (1.5 µg/mL) on a series of MacConkey agar plates: without supplementation, with cefepime (MFEP, 16 µg/mL), with cefoxitin (MFOX, 32 µg/mL), with imipenem (MIPM, 4 µg/mL), and again without supplementation. Plates without antibiotics are intended for inoculum control, while plates with cefoxitin are intended to select for AmpC and carbapenemase producers, cefepime for ESBL and carbapenemase producers, and imipenem for carbapenemase producers. The volunteer providing the swab showed colonization by an AmpC intrinsic producer (*Citrobacter* spp. and *Pseudomonas* spp.) and ESBL-producing *E. coli* confirmed by phenotypic and genotypic tests.

**TABLE 2** Number of strains, bacterial identification, and ESBL-encoding genes recovered from volunteers' swabs by the targeted (CESBL) and the massive culture (MFEP) approaches

| Volunteer number | Bacterial identification | Enzymes encoded by genes found | Number of strains identified by each technique | |
|---|---|---|---|---|
| | | | CESBL | MFEP |
| 2 | *Escherichia* spp. | CTX-M-1/2, CTX-M-8, and TEM | 2 | 1 |
| | | CTX-M-8 and TEM | 0 | 1 |
| 3 | *Escherichia* spp. | CTX-M-1/2 | 2 | 2 |
| | | CTX-M-1/2 and CTX-M-8 | 2 | 0 |
| | *Klebsiella* spp. | CTX-M-1/2, CTX-M-8, TEM, and SHV | 1 | 1 |
| | | CTX-M-1/2, TEM, and SHV | 1 | 1 |
| 9 | *Escherichia* spp. | CTX-M-1/2, CTX-M-8, and TEM | 1 | 2 |
| | | CTX-M-1/2, CTX-M-9, and TEM | 0 | 2 |
| | | CTX-M-1/2 and TEM | 3 | 5 |
| 11 | *Escherichia* spp. | CTX-M-1/2 and CTX-M-8 | 1 | 0 |
| | | CTX-M-1/2, CTX-M-8, and TEM | 0 | 3 |
| | | CTX-M-1/2 and TEM | 1 | 2 |
| | | CTX-M-8 and TEM | 2 | 0 |
| 14 | *Escherichia* spp. | CTX-M-1/2 | 1 | 0 |
| | | CTX-M-1/2, CTX-M-8, and CTX-M-9 | 1 | 3 |
| | | CTX-M-1/2 and CTX-M-9 | 0 | 1 |
| 16 | *Citrobacter* spp. | CTX-M-1/2 and CTX-M-8 | 2 | 2 |
| 18 | *Escherichia* spp. | CTX-M-1/2 and CTX-M-8 | 3 | 1 |
| | | CTX-M-1/2, CTX-M-8, and TEM | 2 | 1 |
| | | CTX-M-1/2 and SHV | 0 | 1 |
| | | CTX-M-8 | 1 | 0 |
| 19 | *Providencia* spp. | CTX-M-1/2, CTX-M-8, and TEM | 0 | 2 |
| | | CTX-M-1/2 and TEM | 3 | 1 |
| | | TEM | 0 | 1 |
| 21 | *Escherichia* spp. | CTX-M-1/2 and CTX-M-8 | 1 | 2 |
| | | CTX-M-1/2, CTX-M-8, and SHV | 1 | 2 |
| 29 | *Escherichia* spp. | CTX-M-1/2 and CTX-M-8 | 6 | 4 |
| | | Total | 37 | 41 |

higher proportion of Enterobacteriaceae (88%) and ESBL producers (67%) compared to CESBL (60% and 37%, respectively). Additionally, MFEP resulted in fewer ESBL-negative Enterobacteriaceae (13 vs 22) and fewer PCR-negative *Acinetobacter* spp. and *Pseudomonas* spp. (3 and 2, respectively) compared to CESBL (17 and 18; Tables 3 and 4). Despite the limited sample size, the diagnostic indicators showed that both methods had 100% sensitivity, but the massive culture method had superior specificity (47% vs 20%) and overall accuracy (68% vs 52%; Table 1). Due to the limited sample size, the observed differences did not reach statistical significance. Both methods showed 100% sensitivity, but conventional culture (MFEP) demonstrated higher specificity (47% vs 20%) and accuracy (68% vs 52%) compared to the CESBL method (Table 1).

The method's greatest strength lies in its enhanced capacity to capture the diversity of ESBL-producing Enterobacteriaceae profiles when considering both microbial identification and beta-lactamase content. Even with a limited number of clinical samples, we identified considerable genetic diversity among ESBL, pAmpC, and NDM carbapenemase producers, highlighting the dynamic nature of gastrointestinal colonization. The strategy effectively captures this complexity using a single specimen and avoids the need to distribute material across multiple selective media, minimizing bias in the evaluation of resistant Enterobacteriaceae colonization.

**TABLE 3** Number and identification of enterobacteria negative for the ESBL production phenotype that were recovered from volunteers' swabs by the targeted (CESBL) and the massive culture (MFEP) approaches

| Volunteer number | Bacterial identification | Number of strains identified by each technique | |
|---|---|---|---|
| | | CESBL | MFEP |
| 7 | *Enterobacter* spp. | 0 | 1 |
| | *Hafnia* spp. | 3 | 4 |
| 10 | *Escherichia* spp. | 4 | 0 |
| 11 | *Enterobacter* spp. | 0 | 1 |
| 15 | *Citrobacter* spp. | 4 | 0 |
| | *Enterobacter* spp. | 2 | 3 |
| 18 | *Escherichia* spp. | 1 | 0 |
| 25 | *Citrobacter* spp. | 2 | 0 |
| | *Enterobacter* spp. | 2 | 0 |
| 28 | *Citrobacter* spp. | 1 | 0 |
| 30 | *Citrobacter* spp. | 1 | 0 |
| | *Escherichia* spp. | 2 | 4 |
| | Total | 22 | 13 |

Further investigation of colonies grown on MFOX and MIPM plates revealed that 11 swabs yielded growth under cefoxitin pressure (39 isolates total). Twenty-six were intrinsic AmpC producers (*Pseudomonas* spp., *Enterobacter* spp., *Citrobacter* spp., and *Acinetobacter* spp.), while 10 *E. coli* and 3 *K. pneumoniae* isolates from four volunteers were screened for pAmpC genes. Among them, four *E. coli* isolates from volunteer 18 harbored $bla_{CMY-like}$, while four *E. coli* from volunteer 30 carried $bla_{NDM-like}$, and two from volunteer 2 were negative for pAmpC but positive for $bla_{TEM-like}$, $bla_{CTX-M-8}$, and $bla_{CTX-M1/2-like}$ genes, matching the profile of isolates from MFEP for the same individual (Table 2). The three *K. pneumoniae* isolates from volunteer 3 did not carry pAmpC genes

**TABLE 4** Number of *Pseudomonas* and *Acinetobacter* spp. negative for ESBL-encoding genes surveyed, and bacteria in which ESBL production is irrelevant, recovered from volunteers' swabs by the targeted (CESBL) and massive culture (MFEP) approaches

| | Volunteer number | Bacterial identification | Number of strains identified by each technique | |
|---|---|---|---|---|
| | | | CESBL | MFEP |
| Strains tested for ESBL production by PCR only, with negative results | 4 | *Pseudomonas* spp. | 5 | 0 |
| | 13 | *Acinetobacter* spp. | 2 | 0 |
| | 15 | *Pseudomonas* spp. | 1 | 0 |
| | 16 | *Pseudomonas* spp. | 5 | 0 |
| | 17 | *Acinetobacter* spp. | 1 | 0 |
| | | *Pseudomonas* spp. | 2 | 0 |
| | 18 | *Acinetobacter* spp. | 1 | 0 |
| | 20 | *Acinetobacter* spp. | 9 | 3 |
| | 25 | *Acinetobacter* spp. | 2 | 0 |
| | 27 | *Acinetobacter* spp. | 2 | 0 |
| | | *Pseudomonas* spp. | 0 | 2 |
| | 28 | *Pseudomonas* spp. | 3 | 0 |
| | 30 | *Pseudomonas* spp. | 2 | 0 |
| Strains not tested for ESBL production | 1 | *Enterococcus* spp. | 2 | 0 |
| | 16 | *Achromobacter* sp. | 0 | 1 |
| | 17 | *Ochrobactrum* sp. | 0 | 1 |
| | 21 | *Empedobacter* sp. | 1 | 0 |
| | 27 | *Stenotrophomonas* spp. | 1 | 0 |
| | | Total | 39 | 7 |

but were positive for $bla_{SHV}$, $bla_{TEM}$, and $bla_{CTX-M1/2-like}$ genes, a genotype also found among MFEP isolates from the same individual (Table 2).

Growth on imipenem-supplemented plates (MIPM) was observed in three swabs (12 isolates: 6 *E. coli*, 2 *Enterobacter* spp., 2 *Klebsiella* spp., and 2 *Stenotrophomonas* spp.). Only *E. coli* from volunteer 30 exhibited acquired carbapenemase activity (EDTA-inhibited) and tested positive for $bla_{NDM}$.

The two-step design—enrichment of beta-lactamase producers followed by replica plating—allowed not only for discrimination among beta-lactamase types but also for exclusion of intrinsic AmpC producers that often yield false positives in commercial media, which likely explains the higher specificity of our method. Additionally, the strategy enabled the identification of diverse ESBL, pAmpC, and carbapenemase-producing strains within the same sample, revealing colonization patterns that would be missed using conventional targeted media.

In this context, developing approaches capable of identifying multiple resistance mechanisms directly from rectal swabs represents a significant advance in microbiological surveillance. Although the present study focused on intestinal colonization, the method's design suggests that it could also be adapted to investigate beta-lactamase producers in other complex sample types such as urine, blood, soil, or aquatic matrices. Such techniques expand our ability to analyze antimicrobial resistance across both clinical and environmental settings, where microbial interactions and horizontal gene transfer play central roles.

Nonetheless, the method's main limitation is the additional day required for processing, which may preclude its use for rapid clinical decision-making or infection control. Therefore, its most appropriate application is in central or research laboratories rather than in frontline diagnostics.

Although not intended for routine clinical diagnostics, our findings suggest that the massive culture approach holds great value for studies investigating intestinal colonization by multidrug-resistant Enterobacteriaceae. It may encourage broader adoption for studying resistance determinants in clinical and environmental samples with potential adaptations for other antimicrobial classes.

## ACKNOWLEDGMENTS

We thank the volunteers who provided the rectal swab samples that made this study possible. We are also grateful for the sponsors Conselho Nacional de Desenvolvimento Científico e Tecnológico (CNPq), Fundação Carlos Chagas Filho de Amparo à Pesquisa do Estado do Rio de Janeiro (FAPERJ); Instituto Nacional de Pesquisa em Resistência Antimicrobiana; and Coordenação de Aperfeiçoamento de Pessoal de Nível Superior. These funding agencies had no role in the design or execution of the study; data collection, management, analysis, or interpretation of the data; preparation, review, or approval of the manuscript; or the decision to submit it for publication.

This work was supported by Conselho Nacional de Desenvolvimento Científico e Tecnológico (CNPq; process numbers 12951/2023-0, 408725/2022-2); by Fundação Carlos Chagas Filho de Amparo à Pesquisa do Estado do Rio de Janeiro (FAPERJ; process numbers, E-26/200.795/2019, E-26/211.554/2019, E-26/201.191/2021, E-26/211.351/2021); by INPRA—Instituto Nacional de Pesquisa em Resistência Antimicrobiana—Brazil (INCT/CNPq: 465718/2014-0); by Coordenação de Aperfeiçoamento de Pessoal de Nível Superior—Brazil [CAPES, financing code 001].

G.T.R.: investigation, methodology, data curation, formal analysis, supervision, writing—original draft, and writing—review and editing. V.O.C.: investigation, methodology, data curation, and writing—review and editing. S.V.M.S.: investigation, methodology, and writing—review and editing. J.B.G.: investigation, methodology, and writing—review and editing. N.Z.C.S.: investigation, supervision, and writing—review and editing. M.G.: supervision and writing—review and editing. R.C.P.: conceptualization, data curation, formal analysis, supervision, and writing—review and editing.

## AUTHOR AFFILIATIONS

[1]Laboratório de Investigação em Microbiologia Médica (LIMM), Instituto de Microbiologia Paulo de Góes, Universidade Federal do Rio de Janeiro, Rio de Janeiro, Brazil

[2]Departamento de Saneamento e Saúde Ambiental, Escola Nacional de Saúde Pública—ENSP, Fiocruz, Rio de Janeiro, Rio de Janeiro, Brazil

[3]Departamento de Medicina Clínica, Universidade Federal Fluminense—UFF, Niterói, Rio de Janeiro, Brazil

[4]Complexo Hospitalar de Niterói—CHN, Niterói, Rio de Janeiro, Brazil

[5]Faculdade de Medicina, Centro de Ciências da Saúde - CCS, Universidade Federal do Rio de Janeiro - UFRJ, Rio de Janeiro, Rio de Janeiro, Brazil

[6]Departamento de Hidrobiologia, Universidade Federal de São Carlos, São Carlos, São Paulo, Brazil

## AUTHOR ORCIDs

Renata Cristina Picão http://orcid.org/0000-0001-6507-0905

## FUNDING

| Funder | Grant(s) | Author(s) |
|---|---|---|
| Conselho Nacional de Desenvolvimento Científico e Tecnológico | 12951/2023-0, 408725/2022-2, 465718/2014-0 | Renata Cristina Picão |
| Fundação Carlos Chagas Filho de Amparo à Pesquisa do Estado do Rio de Janeiro | E-26/200.795/2019, E-26/211.554/2019, E-26/201.191/2021, E-26/211.351/2021 | Renata Cristina Picão |
| Coordenação de Aperfeiçoamento de Pessoal de Nível Superior | financing code 001 | Renata Cristina Picão |

## AUTHOR CONTRIBUTIONS

Gabriel Taddeucci-Rocha, Data curation, Formal analysis, Investigation, Methodology, Supervision, Writing – original draft, Writing – review and editing | Victoria de Oliveira Costa, Data curation, Investigation, Methodology, Writing – review and editing | Sarah Vitória Martins da Silva, Investigation, Methodology, Writing – review and editing | Jéssica Britto Gonçalves, Investigation, Methodology, Writing – review and editing | Natalia Chilinque Zambão da Silva, Investigation, Supervision, Writing – review and editing | Marcia Maiolino Garnica, Supervision, Writing – review and editing | Renata Cristina Picão, Conceptualization, Data curation, Formal analysis, Funding acquisition, Project administration, Writing – original draft, Writing – review and editing

## DATA AVAILABILITY

All data supporting the findings of this study are available as Supplementary Material.

## ETHICS APPROVAL

This study was approved by the Human Research Ethics Committee (protocol number 46762621.2.0000.5455), and each participant signed an informed consent form.

## ADDITIONAL FILES

The following material is available online.

## Supplemental Material

**Supplemental material (Spectrum00157-25-s0001.docx).** Fig. S1; detailed rationale and protocol for screening AmpC-, ESBL-, and carbapenemase-producing gram-negative bacilli using a massive culture-based approach.

**Supplemental data (Spectrum00157-25-s0002.xlsx).** Results for clinical specimens and bacterial isolates analyzed.

**Supplemental video (Spectrum00157-25-s0003.mov).** Detailed visualization of the massive culture procedure.

## Open Peer Review

**PEER REVIEW HISTORY (review-history.pdf).** An accounting of the reviewer comments and feedback.

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
