## [Reviewer comments · Microbiology Spectrum]

Microbiology Spectrum

Massive culture-based approach for the screening of AmpC-, ESBL- and carbapenemase producers from rectal swabs

Gabriel Taddeucci-Rocha, Victoria Costa, Sarah Martins da Silva, Jéssica Gonçalves, Natalia Zambão da Silva, Marcia Garnica, and Renata Picão

Corresponding Author(s): Renata Picão, Universidade Federal do Rio de Janeiro

Review Timeline:

Submission Date:	January 17, 2025
Editorial Decision:	February 25, 2025
Revision Received:	April 28, 2025
Accepted:	May 20, 2025

Editor: Sadjia Bekal

Reviewer(s): Disclosure of reviewer identity is with reference to reviewer comments included in decision letter(s). The following individuals involved in review of your submission have agreed to reveal their identity: Vinay Modgil (Reviewer #2); Neha Khaware (Reviewer #3)

Transaction Report:

DOI: <https://doi.org/10.1128/spectrum.00157-25>

Re: Spectrum00157-25 (**Massive culture-based approach for the screening of AmpC-, ESBL- and carbapenemase producers from complex samples**)

Dear Prof. Renata Cristina Picão:

Thank you for the privilege of reviewing your work. Below you will find my comments, instructions from the Spectrum editorial office, and the reviewer comments.

Revision Guidelines

Sincerely,
Sadjia Bekal
Editor
Microbiology Spectrum

Reviewer #2 (Comments for the Author):

Comments;

The paper outlines a novel extensive culture method for identifying and distinguishing beta-lactamase producers (AmpC, ESBL, and carbapenemases) from intricate materials, particularly rectal swabs. The method demonstrates potential for enhanced selectivity and an expanded detection range relative to a conventional chromogenic agar technique. Nevertheless, the text need

substantial modification before to publication. The absence of a comprehensive methodology and inadequate data presentation constitute significant deficiencies.

Questions;

1. The Materials and Methods section is deficient in essential information required for reproducibility. Numerous elements of the protocol are inadequately detailed, encompassing reagent sources, precise quantities, incubation durations, PCR parameters, and primer sequences. This deficiency in detail considerably undermines the evaluation of the study's validity and reliability.
2. The inclusion of raw data in supplementary material would also be extremely beneficial.
3. The paper references 25 rectal swabs, although the rationale for this sample size is not articulated. Better analysis must be performed to evaluate the sufficiency of the sample size.
4. The manuscript indicates higher specificity and improved accuracy; however, it fails to include real values for sensitivity, specificity, positive predictive value (PPV), and negative predictive value (NPV), together with their corresponding confidence intervals and p-values. The statistical methodology must be clarified with clarity.
5. Please provide complete and detailed descriptions of all methods used, including specific reagent sources, catalog numbers, concentrations, incubation times, PCR conditions (annealing temperature, extension time, etc.), and primer sequences used for PCR.
6. Tables S1-S4 should be presented as tables within the main body of the manuscript.
7. Explain the rationale for choosing a sample size of 25 rectal swabs.
8. The discussion focuses heavily on the technical aspects of the method. Expand on the potential clinical impact and implications of this approach.

Based on these concerns, I recommend that the manuscript be significantly revised before it can be considered for publication. Addressing the points raised above, particularly the lack of methodological detail and the presentation of complete data, is essential. The inclusion of a power analysis and detailed statistical analysis would strengthen the manuscript.

Major Revisions Required.

Reviewer #3 (Comments for the Author):

1. Repetition in Importance and Abstract: The abstract and importance sections currently exhibit significant redundancy. Reducing repetitive content and focusing more on articulating the problem statement within the importance section is recommended to enhance clarity and impact. Specifically, the importance section should emphasize the research gap and its significance, rather than elaborating extensively on the methodology. This adjustment will better contextualize the study's objectives and relevance.
2. Comparison Between CESBL and MFEP: The manuscript provides a detailed evaluation and comparison of the results obtained using the commercial CESBL method and the proposed MFEP method. It is clearly demonstrated that MFEP outperforms CESBL, offering higher accuracy as validated by PCR analysis. However, to ensure the robustness and reliability of the findings, it is crucial to include the PCR protocol and detailed results. This addition will allow reviewers to thoroughly assess the performance of MFEP, lending greater credibility to the conclusions drawn in the study.
3. The current title, which includes "from complex samples," may not accurately reflect the scope of the study, as the evaluation has been conducted solely with rectal swabs. A more precise and representative title could be: "Massive culture-based approach for the screening of AmpC-, ESBL-, and carbapenemase producers from rectal swabs." However, the authors could note within the manuscript that the proposed method has the potential to be applied to other sample types, thereby expanding its scope and utility in future studies.
4. The resolution of the culture plates presented in Figure 2 is suboptimal and appears compromised. Additionally, it is unclear why the 10^6 CFU/mL inoculum exhibits less growth compared to the 10^5 CFU/mL inoculum.

Comments;

The paper outlines a novel extensive culture method for identifying and distinguishing beta-lactamase producers (AmpC, ESBL, and carbapenemases) from intricate materials, particularly rectal swabs. The method demonstrates potential for enhanced selectivity and an expanded detection range relative to a conventional chromogenic agar technique.

Nevertheless, the text need substantial modification before to publication. The absence of a comprehensive methodology and inadequate data presentation constitute significant deficiencies.

Questions;

1. The Materials and Methods section is deficient in essential information required for reproducibility. Numerous elements of the protocol are inadequately detailed, encompassing reagent sources, precise quantities, incubation durations, PCR parameters, and primer sequences. This deficiency in detail considerably undermines the evaluation of the study's validity and reliability.
2. The inclusion of raw data in supplementary material would also be extremely beneficial.
3. The paper references 25 rectal swabs, although the rationale for this sample size is not articulated. Better analysis must be performed to evaluate the sufficiency of the sample size.
4. The manuscript indicates higher specificity and improved accuracy; however, it fails to include real values for sensitivity, specificity, positive predictive value (PPV), and negative predictive value (NPV), together with their corresponding confidence intervals and p-values. The statistical methodology must be clarified with clarity.
5. Please provide complete and detailed descriptions of all methods used, including specific reagent sources, catalog numbers, concentrations, incubation times, PCR conditions (annealing temperature, extension time, etc.), and primer sequences used for PCR.
6. Tables S1-S4 should be presented as tables within the main body of the manuscript.
7. Explain the rationale for choosing a sample size of 25 rectal swabs.
8. The discussion focuses heavily on the technical aspects of the method. Expand on the potential clinical impact and implications of this approach.

Based on these concerns, I recommend that the manuscript be significantly revised before it can be considered for publication. Addressing the points raised above, particularly the lack of methodological detail and the presentation of complete data, is essential. The inclusion of a power analysis and detailed statistical analysis would strengthen the manuscript.

Major Revisions Required.

1. **Repetition in Importance and Abstract:** The abstract and importance sections currently exhibit significant redundancy. To enhance clarity and impact, reducing repetitive content and focusing more on articulating the problem statement within the importance section is recommended. Specifically, the importance section should emphasize the research gap and its significance, rather than elaborating extensively on the methodology. This adjustment will better contextualize the study's objectives and relevance.
2. **Comparison Between CESBL and MFEP:** The manuscript provides a detailed evaluation and comparison of the results obtained using the commercial CESBL method and the proposed MFEP method. It is clearly demonstrated that MFEP outperforms CESBL, offering higher accuracy as validated by PCR analysis. However, to ensure the robustness and reliability of the findings, it is crucial to include the PCR protocol and detailed results. This addition will allow reviewers to thoroughly assess the performance of MFEP, lending greater credibility to the conclusions drawn in the study.
3. **Title:** The current title, which includes "from complex samples," may not accurately reflect the scope of the study, as the evaluation has been conducted solely with rectal swabs. A more precise and representative title could be: **"Massive culture-based approach for the screening of AmpC-, ESBL-, and carbapenemase producers from rectal swabs."** However, the authors could note within the manuscript that the proposed method has the potential to be applied to other sample types, thereby expanding its scope and utility in future studies.
4. **Figure 2:** The resolution of the culture plates presented in Figure 2 is suboptimal and appears compromised. Additionally, it is unclear why the 10^6 CFU/mL inoculum exhibits less growth compared to the 10^5 CFU/mL inoculum

Massive culture-based approach for the screening of AmpC-, ESBL- and carbapenemase producers from rectal swabs

Gabriel Taddeucci-Rocha^{1,2}, Victoria de Oliveira Costa¹, Sarah Vitória Martins da Silva¹,
Jéssica Britto Gonçalves¹, Natalia Chilingue Zambão da Silva^{3,4}, Marcia Maiolino
Garnica^{4,6}, Renata Cristina Picão^{1,6*}

Response to Reviewers

The authors thank the editor and reviewers that dedicated their efforts and time to evaluate our manuscript.

Please find below the point by point response to comments provided.

Sincerely,

The authors

COMMENTS FROM THE EDITOR AND/OR REVIEWERS

REVIEWER #1 (Comments for the Author):

The paper outlines a novel extensive culture method for identifying and distinguishing beta-lactamase producers (AmpC, ESBL, and carbapenemases) from intricate materials, particularly rectal swabs. The method demonstrates potential for enhanced selectivity and an expanded detection range relative to a conventional chromogenic agar technique. Nevertheless, the text need substantial modification before to publication. The absence of a comprehensive methodology and inadequate data presentation constitute significant deficiencies.

We sincerely appreciate your valuable feedback and constructive suggestions regarding our manuscript. We have carefully addressed all the points raised and made the necessary revisions to improve the clarity, methodology, and data presentation. Below, we outline the key modifications implemented in response to your comments.

1. The Materials and Methods section is deficient in essential information required for reproducibility. Numerous elements of the protocol are inadequately detailed, encompassing reagent sources, precise quantities, incubation durations, PCR parameters, and primer sequences. This deficiency in detail considerably undermines the evaluation of the study's validity and reliability.

We have added all requested methodological details to the revised manuscript, including: reagent sources, exact quantities, incubation conditions, complete PCR parameters, and primer sequences. For established protocols, we referenced prior publications to maintain conciseness while ensuring reproducibility. The Methods section now provides complete information to replicate this study.

2. The inclusion of raw data in supplementary material would also be extremely beneficial.

Thank you for your valuable suggestion. In response to your comment regarding the inclusion of raw data, we would like to inform you that we have now provided a supplementary table containing complete datasets for both clinical specimens and bacterial isolates.

3. The paper references 25 rectal swabs, although the rationale for this sample size is not articulated. Better analysis must be performed to evaluate the sufficiency of the sample size.

We appreciate your comment regarding the sample size. As requested, we have added an explanation for the number of analyzed samples in the Materials and Methods section. As detailed in the revised manuscript, we established a 3-month collection period in the study timeline. During this period, we were able to recruit 30 volunteers, of which 25 remained in the study after viability analysis of the collected swabs.

4. The manuscript indicates higher specificity and improved accuracy; however, it fails to include real values for sensitivity, specificity, positive predictive value (PPV), and negative predictive value (NPV), together with their corresponding confidence intervals and p-values. The statistical methodology must be clarified with clarity.

We thank the reviewer for highlighting this important aspect regarding the statistical analyses. In response, we have performed the appropriate statistical tests and included the confidence intervals and p-values. These results have been properly incorporated into Table 1 of the revised manuscript. Additionally, we have included a specific subsection on statistical analysis in the Methods section. Furthermore, we have expanded the Discussion section to include an interpretive analysis of these statistical results. We believe these modifications have contributed to enhancing both the methodological rigor and practical relevance of our findings.

5. Please provide complete and detailed descriptions of all methods used, including specific reagent sources, catalog numbers, concentrations, incubation times, PCR conditions (annealing temperature, extension time, etc.), and primer sequences used for PCR.

We have added all requested methodological details to the revised manuscript, including: reagent sources, exact quantities, incubation conditions, complete PCR parameters, and primer sequences. For established protocols, we referenced prior publications to maintain conciseness while ensuring reproducibility. The Methods section now provides complete information to replicate this study.

6. Tables S1-S4 should be presented as tables within the main body of the manuscript.

Thank you for the suggestion. It has been accepted, and Tables S1–S4 have been included in the main body of the manuscript.

7. Explain the rationale for choosing a sample size of 25 rectal swabs.

As clarified above, we have added an explanation for the number of analyzed samples in the Materials and Methods section. As detailed in the revised manuscript, we established a 3-month collection period in the study timeline. During this period, we were able to recruit 30 volunteers, of which 25 remained in the study after viability analysis of the collected swabs.

8. The discussion focuses heavily on the technical aspects of the method. Expand on the potential clinical impact and implications of this approach.

We appreciate your comment. We have made significant changes to the Results and Discussion sections, incorporating your suggestions. We have now expanded the discussion on the potential clinical impact and implications of this new approach.

REVIEWER #2 (Comments for the Author):

1. Repetition in Importance and Abstract: The abstract and importance sections currently exhibit significant redundancy. Reducing repetitive content and focusing more on articulating the problem statement within the importance section is recommended to enhance clarity and impact. Specifically, the importance section should emphasize the research gap and its significance, rather than elaborating extensively on the methodology. This adjustment will better contextualize the study's objectives and relevance.

We sincerely appreciate the reviewer's insightful suggestion regarding the redundancy between the Abstract and Importance sections. We fully agree with this observation and have carefully revised both sections to eliminate repetitive content.

2. Comparison Between CESBL and MFEP: The manuscript provides a detailed evaluation and comparison of the results obtained using the commercial CESBL method and the proposed MFEP method. It is clearly demonstrated that MFEP outperforms CESBL, offering higher accuracy as validated by PCR analysis. However, to ensure the robustness and reliability of the findings, it is crucial to include the PCR protocol and detailed results. This addition will allow reviewers to thoroughly assess the performance of MFEP, lending greater credibility to the conclusions drawn in the study.

We sincerely appreciate your valuable feedback and have included the requested PCR protocol and detailed results in the revised manuscript.

3. The current title, which includes "from complex samples," may not accurately reflect the scope of the study, as the evaluation has been conducted solely with rectal swabs. A more precise and representative title could be: "Massive culture-based approach for the screening of AmpC-, ESBL-, and carbapenemase producers from rectal swabs." However, the authors could note within the manuscript that the proposed method has the potential to be applied to other sample types, thereby expanding its scope and utility in future studies.

We sincerely appreciate your suggestion and fully agree that the revised title, "Massive culture-based approach for the screening of AmpC-, ESBL-, and carbapenemase producers from rectal swabs," better reflects the study's scope, and we have updated it accordingly, while also adding a note in the Discussion section to emphasize the method's potential applicability to other sample types in future studies, further expanding its utility.

4. The resolution of the culture plates presented in Figure 2 is suboptimal and appears compromised. Additionally, it is unclear why the 10^6 CFU/mL inoculum exhibits less growth compared to the 10^5 CFU/mL inoculum

We appreciate your observation and the opportunity to clarify these points. Indeed, there was an error in the initial concentrations indicated, which has now been corrected in both the text and the figure. Regarding the methodology, we started with an initial inoculum standardized to the 0.5 McFarland scale, which was subsequently diluted to concentrations of 10^{-3} and 10^{-4} , resulting in bacterial loads of 10^5 and 10^4 CFU/mL, respectively.

Re: Spectrum00157-25R1 (**Massive culture-based approach for the screening of AmpC-, ESBL- and carbapenemase producers from rectal swabs**)

Dear Prof. Renata Cristina Picão:

Your manuscript has been accepted, and I am forwarding it to the ASM production staff for publication. Your paper will first be checked to make sure all elements meet the technical requirements. ASM staff will contact you if anything needs to be revised before copyediting and production can begin. Otherwise, you will be notified when your proofs are ready to be viewed.

Sincerely,
Sadjia Bekal
Editor
Microbiology Spectrum

Reviewer #2 (Comments for the Author):

Accepted.

The author has provided satisfactory and well-reasoned responses to all the queries raised by the reviewers. The manuscript demonstrates scientific merit and clarity in addressing the concerns. It is recommended for acceptance subject to minor grammatical corrections and language polishing.